# Fuzzy-Logic Dijkstra-Based Energy-Efficient Algorithm for Data Transmission in WSNs

**DOI:** 10.3390/s19051040

**Published:** 2019-02-28

**Authors:** Madiha Razzaq, Seokjoo Shin

**Affiliations:** Department of Computer Engineering, Chosun University, Gwangju 61452, Korea; madiha_razzaq@ymail.com

**Keywords:** energy-efficiency, weighted sum method, Dijkstra’s algorithm, fuzzy membership function, intra-cluster communication cost

## Abstract

In wireless sensor networks, clustering routing algorithms have been widely used owing to their high energy-efficiency and scalability. In clustering schemes, the nodes are organized in the form of clusters, and each cluster is governed by a cluster head. Once the cluster heads are selected, they form a backbone network to periodically collect, aggregate, and forward data to the base station using minimum energy (cost) routing. This approach significantly improves the network lifetime. Therefore, a new cluster head selection method that uses a weighted sum method to calculate the weight of each node in the cluster and compare it with the standard weight of that particular cluster is proposed in this paper. The node with a weight closest to the standard cluster weight becomes the cluster head. This technique balances the load distribution and selects the nodes with highest residual energy in the network. Additionally, a data routing scheme is proposed to determine an energy-efficient path from the source to the destination node. This algorithm assigns a weight function to each link on the basis of a fuzzy membership function and intra-cluster communication cost within a cluster. As a result, a minimum weight path is selected using Dijkstra’s algorithm that improves the energy efficiency of the overall system. The experimental results show that the proposed algorithm shows better performance than some existing representative methods in the aspects of energy consumption, network lifetime, and system throughput.

## 1. Introduction

Wireless sensor networks (WSNs) have received great attention owing to their wide usage in potential applications in numerous fields [1]. These networks comprise a large number of sensing devices known as sensor nodes. These sensor nodes monitor the target area of deployment and collectively transmit the monitored information to the base station (BS). Additionally, the sensors have limited power, sensing, communication, processing, and memory resources. The small size and economical cost of these sensors contribute to their limited resources. The size of a sensor node is application specific. It may vary from a microscopic particle, e.g., a tooth sensor, to a brick, e.g., a weather station. Their cost also differs with the degree of the required sensor capabilities, e.g., sensors equipped with simple hardware design have a low cost and those with complex hardware design have a higher cost. Therefore, changes in the footprint and price of the sensor have a significant impact on its resource limitations [2].

Sensor nodes are primarily battery operated devices; therefore, the battery is the most critical and precious resource of a sensor. It has considerable effect on the overall lifetime of a WSN. Energy depletion of sensor node batteries results in coverage holes in the network, i.e., such nodes become non-operational and cease to participate in information sensing or data collection. Moreover, this may lead to connectivity loss in different parts of the network because the communication routes may be damaged owing to node failure events [3]. Therefore, the energy consumption of the sensor node must be correctly managed to increase the network lifetime and functionality for a rational amount of time. The lifetime of a network is defined in the literature as the time duration elapsed until the first node in the network dies, i.e., depletes its energy [4]. 

Network topology control provides an efficient approach for addressing such challenges in the WSN [5]. Topology control is defined as a set of techniques or methods that can transform an underlying network structure in an efficient manner, thereby maximizing system performance or reducing transmission cost [6]. Several techniques of topology control have been proposed for WSNs; however, clustering has proven to be the most efficient and widely used scheme for managing network structure. Moreover, cluster-based routing mechanisms have been presented in the Refs. [5,6,7]; these mechanisms comprise two phases: the set-up phase and the steady state phase. In the set-up phase, all the sensor nodes are organized in the form of clusters. In each cluster, a cluster head (CH) selection algorithm is executed to elect a node as the leader of the corresponding cluster and perform related tasks. Therefore, each cluster consists of member nodes and a CH. After the CH selection, the steady-state phase starts and the CH aggregates the data received from surrounding member nodes and transmits it to the BS directly (single-hop communication) or through other CH nodes (multi-hop communication) as illustrated in Figure 1. 

On the contrary, multi-hop communication within a cluster can reduce the number of communication links and avoid the communication congestion compared to direct communication. This is owing to the fact that the CH has to communicate with more member nodes simultaneously. Additionally, multi-hop communication can enable member nodes to assist the CH in sharing the task of data aggregation and minimize the energy consumption of CHs. This leads to the enhanced lifetime of the underlying network. Cluster heads gather the collected information from all the member nodes. After the data aggregation process, the CHs forward the data to the BS in multi-hop manner to perform the data upload [8]. In cluster-based network systems, the nodes are randomly deployed with different distances and energy consumptions. The formed clusters with equal number of nodes always lead to uneven energy consumption in the network. Therefore, unequal clustering techniques are usually adopted to ensure load balancing in the system. 

While designing a cluster-based routing algorithm, selection of CH should be performed efficiently because a CH executes energy intensive functions in a cluster such as data receiving, aggregation, and transmission to the BS. These functions consume more amounts of energy compared to the data sensing tasks performed by the surrounding nodes. Furthermore, the distance between the selected node and the BS should be relatively small to increase the energy conservation in the network. Therefore, to elect the appropriate CH, factors such as node residual energy and the distance of the node to BS should be considered to ensure selection of an efficient sensor node among the other cluster nodes. This study proposes an efficient CH selection method based on a weighted sum method to ensure the selection of an energy rich node in the whole network for each cluster.

Moreover, the maximum amount of energy is consumed in the steady state phase wherein data is transmitted from the sensor nodes to the main access point, i.e., the BS in the network [6]. This leads to the need for an energy efficient routing algorithm that can solve the data routing problem of energy-constrained nodes. In WSNs, a CH consumes much more energy than the member nodes, owing to its particular responsibilities. However, during the intra-cluster multi-hop communication, the sensor nodes located closer to the CH are required to receive and forward data more frequently compared to the farther nodes, which may lead to more energy depletion. This extra dissipation of energy minimizes the lifetime of the corresponding relay nodes and induces an energy imbalance in the whole network system [9]. In WSNs, while monitoring the environment, if the data forwarding nodes die owing to insufficient residual energy during each routing round, then the further associated nodes have to rediscover the routing path and reestablish the communication link, which will increase the energy consumption in the system. This behavior destructs the stability of the communication and leads to inefficient network performance. Therefore, the residual energy of the relay nodes within a cluster has a great influence on the network energy efficiency, and it plays a significant role in the data transmission process. Furthermore, selecting an efficient path from the source to the destination node also helps in minimizing the network energy consumption. In WSNs, the weights or costs of the links (edges) between neighbor nodes are calculated based on certain factors, and are assigned to the edges. This link or edge weight decides the selection of the optimal path comprising consecutive edges for which the overall cost is minimized. Several algorithms are used in this regard such as Floyd–Warshall, Dijkstra’s, and Bellman–Ford techniques [10,11]. These algorithms are diverse in nature owing to the consideration of different processing times and the amount of information required from the nodes, thereby making each of them applicable to a specific routing scenarios. Furthermore, Machine learning (ML) is a significant tool to utilize distributive characteristics of WSNs and improve the network lifetime. Neural networks (NNs), fuzzy logic (FL), evolutionary algorithms (EAs), reinforcement learning (RL), and swarm intelligence (SI) are different ML techniques used in WSNs to overcome the challenges offered by the resource constrained nature of sensor nodes. Fuzzy logic, owing to its potential, has been fully explored in various fields such as signal processing, facial and speech recognition, medical, robotics, marketing, and networking [12,13]. Also, FL is a promising technique for WSNs since it allows the combination and evaluation of several network parameters in an efficient manner. Fuzzy logic is able to improve overall network performance as its execution requirements can be easily supported by sensor nodes. In WSNs, this approach is used in localization techniques, cluster formation, and CH selection process, security, and data routing methods [14].

In this study, a routing scheme with an efficient CH selection technique is proposed to provide a balanced energy consumption in the whole network. Additionally, a weight function is formulated on the basis of a fuzzy membership function. An intra-cluster communication cost function is also proposed to choose an energy-efficient and delay tolerant path from the source to destination node using Dijkstra’s algorithm. The fuzzy membership function chooses the relay node with maximum energy among the other neighboring nodes. On the contrary, the intra-cluster communication cost function ensures to minimize the communication cost within a cluster. As a result, the nodes closer to the CH do not deplete their energies faster. This technique induces a high degree of energy efficiency in the network. The main contributions of this paper are as follows:To formulate a multi-objective weighted sum function that would select an efficient CH node in a cluster and reduce energy consumption imbalance in the network.To address the node isolation issue in the network by presenting the weight function which is influenced by two functions, i.e., a fuzzy membership function and an intra-cluster communication cost function to reduce the energy depletion during data communication from the source to the destination node within a cluster.To conduct a series of experiments to evaluate the performance of the proposed routing scheme and demonstrate that the proposed algorithm performs better than the other existing routing protocols in terms of energy efficiency and network throughput.

The remainder of this paper is organized as follows: Section 2 provides an overview of the related work for the cluster-based energy efficient routing schemes. Section 3 explains the system including the network and energy model. The proposed fuzzy logic Dijkstra-based routing scheme is explained in detail in Section 4. The performance analysis and evaluation and the simulation results are presented in Section 5. The paper concludes in Section 6. 

## 2. Related Work

Several energy-efficient and energy-balanced routing protocols are developed in the literature with the objective of extending the network lifetime of sensor networks and improving the overall system performance. In Ref. [10], low-energy adaptive clustering hierarchy (LEACH), which is the most popular and self-organizing adaptive clustering routing algorithms for WSNs in the early times is proposed. It has the ability to increase the robustness and scalability of dynamic networks to a certain degree. The scheme proposes the generation of a random number between 0 and 1 that helps in electing the CH if the random number is less than a threshold value *T*(*n*), which is defined in Equation (1).
(1)$$T\left(n\right)=\{\begin{array}{c} \frac{p}{1-p\left[r\;mod\left(1/p\right)\right]},\;n\;\in G\\ \;0,\;others \end{array}$$
where *p* is the probability of the node to be selected as the CH during the election phase, *r* indicates the current election round, and *G* denotes the set of nodes that were not selected as CHs in the previous rounds. Each node has equal probability to become the CH. Once the CH is elected, each cluster member is provided a time slot to transmit the data to the CH in a time division multiple access (TDMA) manner. Although LEACH does not entirely take the communication energy consumption into account, and the random selection of the CH causes various deficiencies, it provides a good theoretical basis and design model for subsequent clustering techniques. 

A hybrid and energy-efficient distributed (HEED) clustering approach is presented in Ref. [15]. It initializes the CH selection phase by setting two parameters: primary and secondary. The primary parameter considers the residual energy of nodes and selects the node with highest residual energy as CH, whereas the secondary parameter reflects the nodes’ density to compute the communication cost within the cluster. In HEED, the cluster formation occurs in a distributed manner, and it maintains a good distribution for the CH nodes. Moreover, it helps in improving the network lifetime by reducing the communication energy consumption within the cluster. However, in the cluster formation phase, several messages are broadcasted to exchange the network information, which results in additional energy consumption by the systems. Cluster-based event-driven routing protocol (CERP), in which cluster formation takes place based on different events, is proposed in Ref. [16]. The algorithm considers the distance between the neighboring nodes as a link cost while calculating the shortest path from the source to the sink node. Clusters are formed on the basis of event occurrence; therefore, the routing scheme may encounter less energy intermediate nodes owing to their random deployment. This may lead to the network isolation problem in some areas. 

Furthermore, a Dijkstra-based weighted sum minimization (DWSM) algorithm is proposed in Ref. [17] for wireless mesh networks (WMNs). It introduces a multi-objective function as the link cost between the nodes, which is influenced by the network parameters such as end-to-end delay and path capacity. This scheme analyzes the impact of these two factors on the network performance by minimizing a weight metric sum through Dijkstra’s algorithm. In Ref. [18], hierarchical unequal clustering fuzzy algorithm (HUCFA) is introduced to reduce the energy consumption of the network. It divides the network area into three horizontal layers based on its distance to BS and then splits each layer into grids. Furthermore, it includes a fuzzy-logic-based CH selection scheme which enhances the energy efficiency. An energy saving routing algorithm based on Dijkstra (ESRAD) is proposed in Ref. [11] for WSNs. This scheme considers an evaluation index of the node by considering the energy consumption of information processing at the node and energy consumption due to the transmission between two connected nodes. This index is then used as a link cost through Dijkstra’s algorithm to search the path with the least energy consumption. The main drawback of this scheme is that it does not consider the residual energy of the neighboring nodes, which can result in inappropriate load balancing in the network. Shortest path evaluation using fuzzy logic (SPFL), which uses a fuzzy logic function for data routing in WSNs is proposed in Ref. [19]. It introduces the concept of a pool manager (PM) node that is located near the source node, and has no hard limitations of energy, memory, or computational capabilities. Whenever data needs to be transmitted, a request is sent to the PM by the source node to select the shortest path to the selected receiver node. The PM broadcasts knock messages to all the nodes, and on the basis of time delay of each received reply, it decides the least delay route and informs the source node about it. The scheme ignores the residual energy of the nodes; this affects the overall performance of the entire network system.

Adaptive clustering for mobile wireless networks is proposed in Ref. [20] which takes the multi-hopping ability of nodes into account. It is a distributive clustering approach where clusters are independently controlled and dynamically reconfigured with node mobility. This study organizes the nodes in the form of non-overlapping clusters by considering a trade-off between spatial reuse of communication channel and delay minimization which provides an efficient and stable infrastructure for dynamic radio networks. In Ref. [21], a CH-token infrastructure for multi-hop mobile WSNs is proposed. This scheme considers the radio channel access, code scheduling, and channel reservation parameters for data routing which reduces packet delay performance in the network. In Ref. [22], the author proposes a mechanism for cluster formation based on social interest among nodes for exchanging information along physical and network-related parameters such as available energy, channel quality, and physical proximity of the nodes in machine-to-machine (M2M) communication. This is done by introducing a threshold-based M2M link establishment concept in which an application specific threshold value is assumed to compute the interest of two nodes towards sharing the data. This approach efficiently decreases energy consumption of the internet of things (IoT) devices and eventually results in maximized network lifetime. Similarly, Ref. [23] presents a cluster formation and CH selection mechanism which is influenced by the social, physical and mobility characteristics of the users in public safety networks (PSNs) for supporting device-to-device (D2D) communication. It introduces a multi-objective weight function based on the aforementioned network characteristics and the node/device with minimum weight is selected to perform the duties of CH. This technique enhances the network lifetime and the scalability for increasing the number of devices in PSNs. 

A fuzzy maximum lifetime (FML) algorithm is a routing scheme that proposes a fuzzy membership function for maximizing the network lifetime [24]. It assigns high membership value to the edge having a large amount of residual energy at its starting node. This membership value is then used to compute the weight of the link. Further, through Dijkstra’s algorithm, the minimum weight route with the maximum lifetime is selected for data transmission. This scheme focuses on the residual energy of the source node and neglects the energy consumption of the neighbor nodes and data transmission; this may cause poor performance of the overall network system. In Ref. [25], distributed unequal clustering using fuzzy logic (DUCF) is proposed. It is based on fuzzy logic for unequal clustering networks, which chooses the node’s degree, residual energy, and the average node distance to the BS as the input variables. As a result, the output is the size of the cluster and the elected CH. Distributed unequal clustering using fuzzy logic not only considers a node’s own factor but also the cluster size, which leads to enhanced performance of the whole network. In addition to this, several reinforcement learning techniques have been introduced for improving network lifetime in WSNs [26]. In Ref. [27], a two-stage sleep scheduling approach is introduced based on reinforcement learning for providing area coverage in WSNs. This approach enables sensor nodes to sparsely cover the target area without the need for node location information. This is done by Q-learning algorithm to select a limited number of nodes which can appropriately cover the deployment area. As a result, the network lifetime is improved and the desired area coverage is achieved. In Ref. [28], a learning algorithm is proposed to determine the optimal cell selection which includes a power allocation mechanism. Each user in a two-tier femtocell network is associated with a QoS-aware function which ensures the selection of appropriate cell to connect and maximizes QoS-based performance. A study on energy-saving routing algorithm for WSNs (LEACH-DT) is proposed in Ref. [29]. This scheme introduces a dynamic energy threshold value to reduce the energy consumption during the CH selection and rotation phase. Additionally, it resolves the issue of reallocation of time slots for each round by exchanging the time slot information between the candidate CH and the current CH. In Ref. [30], a study was documented on optimizing the relay nodes by tree-cluster based shortest path (TCBSP) in WSNs. The routing scheme ensures proper connectivity among each cluster member node by organizing the nodes in the form of a tree cluster structure. This method improves the lifespan of the individual nodes in the network because it first aggregates the data in a cluster form and then transmits it to another cluster in the form of a tree. The shortcoming of this scheme is that it does not consider the residual energy of individual nodes while selecting the CH because each node has equal probability of becoming the CH, which may result in the selection of a node with less energy as the CH, and further causes network failure and critical data loss. Table 1 shows a summary of performance characteristics. It summarizes the key features, advantages, and disadvantages of some of the clustering-based routing algorithms discussed above.

Considering the inadequacy of the abovementioned algorithms, a routing algorithm, which is an energy-efficient fuzzy logic Dijkstra-based routing scheme for WSNs, is proposed in this study. In the proposed scheme, the nodes are organized in the form of clusters using a K-means clustering scheme. Each cluster is then assigned a CH node by calculating a weighted sum function for each cluster node and then comparing with the standard weight of the cluster. Once the network is set-up, the data transmission phase starts on the detection of an event. The node near the location of the event senses the information and transmits the data to the final node through an energy-efficient and delay resistant route by considering the neighbor nodes’ residual energy and the cost of intra-cluster communication. This approach avoids unnecessary energy consumption of the relay nodes by load balancing; therefore, it is more helpful for the optimization of energy consumption and extends the lifetime of the network system.

## 3. System Model

### 3.1. Sensor Network Model and Assumptions

In designing the routing protocol, the network model presents the sensor nodes’ operating environment that consists of n homogeneous sensor nodes, i.e., $N_{i}\left(i=1,\;2,\;3,\ldots \ldots ,n\right)$. The sensor nodes are distributed randomly in a target monitoring area of *F × F* dimensions, and they are stationary after deployment. Each node has the ability to sense the events (activity), collect, process, and communicate data within the network. The nodes are equipped with battery energy that is limited and mostly consumed during the data transmission and reception process at its radio transceiver. In traditional WSNs, the nodes are left unattended after deployment; therefore, battery replacement or recharge is not feasible. Data sensing occurs when an event occurs, e.g., rise in the temperature of the environment wherein the nodes are deployed. In the proposed scheme, an event is defined as the variation in the sensed value beyond a certain predefined threshold level. In Figure 2, the car accident represents the occurrence of an event as it will result in an increased environmental value. The cluster member (CM) node located near the event-occurring area will sense the data and report it to the CH. The sensed information is transmitted to the BS through CHs. The BS is located far from the network area, and it does not have limitations regarding computational power, energy, and memory resources. The proposed scheme is a centralized routing algorithm in which the centralized entity is the BS. Additionally, it considers that all the transmissions to the BS are solely through the CH using multi-hop communication, except for the first communication round. Further, in our scheme the CH selection process takes residual energy and intra-cluster communication cost into account for selecting a node as the CH. This results in selection of a head node which is well-located and has a sufficient amount of energy to transmit data from member nodes to the BS.

In this study, the sensor nodes were organized in the form of clusters. Each cluster had its own CH, and it was responsible for data aggregation and transmission to the BS in a multi-hop manner. Nodes belonging to the clusters were referred to as CMs, and the nodes that could be reached by a single-hop transmission using a maximum transmission radius $r_{t}$ were referred to as neighbor nodes. Additionally, only the member node near which the event occurred delivered its sensed data to the CH over multi-hop paths. These multi-hop paths were formed as a result of the next hop choices made on the basis of a weight function, which is discussed in detail in Section 4.4: Data Transmission Phase. This resulted in energy-efficient data packet transmission from the source node (CM) to the destination node (CH). Once data reached the CH, the CH aggregated it and transmitted it to the BS. Figure 2 illustrates the proposed sensor network topology. 

### 3.2. Energy Consumption Model

The energy model used in the proposed scheme is adopted from Ref. [31]. The radio energy dissipation model of a transceiver consists of three primary components: a transmitter, amplifier, and receiver, which are illustrated in Figure 3. 

The energy model formulates the energy expended by the ith sensor node amid the transmission and reception of a p-bit packet to jth sensor node as shown by Equations (2) and (3): (2)$$E_{TX}\left(p,d_{ij}\right)=\left(E_{elec}+\varepsilon_{amp}\right)\cdot p$$
(3)$$E_{RX}\left(p\right)=E_{elec}\cdot p$$
where $d_{ij}$ represents the distance between the ith and the jth node, and $E_{TX}$ and $E_{RX}$ are the transmission and reception energy dissipation, respectively. $E_{elec}$ is a distance-independent entity, and it accounts for the energy consumption by the transceiver circuitry. $\varepsilon_{amp}$ denotes the energy consumption of the transmitter’s amplifier, which is formulated as:(4)$$\varepsilon_{amp}=\{\begin{array}{c} \varepsilon_{fs}\cdot \;d_{ij}{}^{2}\;for\;d_{ij}\leq d_{th}\\ \varepsilon_{mp}\cdot \;d_{ij}{}^{4}\;for\;d_{ij}>d_{th} \end{array}$$
and (5)$$d_{th}=\sqrt{\frac{\varepsilon_{fs}}{\varepsilon_{mp}}}$$
where $d_{th}$ is the threshold distance, which is defined in Equation (5), $\varepsilon_{fs}$ and $\varepsilon_{mp}$ are the propagation models for the amplification energy dissipation in free space and multi-path, respectively. In the free space model, a direct line-of-sight (LOS) path is considered between the transmitter and receiver nodes, whereas a multipath represents the non-line-of-sight (NLOS) signal propagation from various routes at different time intervals after reflection from the ground. 

The amount of energy consumed by a CM in a cluster:$\;E_{CM},$ and the amount of energy required for a CH: $E_{CH},$ are given by Equations (6) and (7), respectively, as: (6)$$E_{CM}=E_{init}-E_{TX}\left(p,d_{ij}\right)$$
(7)$$E_{CH}=E_{init}-E_{std}$$
(8)$$E_{std}=E_{TX}\left(p,d_{ij}\right)+E_{DA}+E_{RX}\left(p\right)$$
where $E_{init}$ represents the initial energy of the sensor node. $E_{std}$ is the standard energy consumption of a node taking part in the CH selection phase, and $E_{DA}$ denotes the energy consumption of the node for the data aggregation process.

## 4. Proposed Fuzzy Logic Dijkstra-based Routing Algorithm

The proposed algorithm comprises four main phases: (1) network deployment, (2) cluster formation, (3) CH selection, and (4) data transmission phase. The phases are explained in detail below.

### 4.1. Network Deployment Phase

This is the initial phase, wherein the network management control messages are exchanged between the BS and sensor nodes. The BS broadcasts the initialization request message ($M_{IRQ}$) to all the sensor nodes in the network field. Once the $M_{IRQ}$ message is received by a sensor node, it responds with an initialization reply message ($M_{IRP}$). This exchange of messages occurs to know the current location and residual energy of the nodes in the deployment area [31]. 

### 4.2. Cluster Formation Phase

In this phase, the sensor nodes are grouped in the form of clusters by using *K*-means clustering algorithm. This algorithm is widely used to group the data into *K* clusters. In the proposed algorithm, the value of *K* is calculated by Equation (9) [31]. As a result, the network domain is divided into *K* clusters with minimum inter-cluster and maximum intra-cluster similarity.
(9)$$C_{K}=\sqrt{\frac{n}{2\pi }}\cdot \frac{F}{d_{BS}{}^{2}}\cdot \;d_{th}$$
where *n* denotes the number of sensor nodes, *F* represents the network domain size, and $d_{BS}{}^{2}$ symbolizes the average node distance from the BS. This clustering algorithm comprises several iterations and steps that are described as follows:

Step I: Calculate the value of $C_{K}$ using Equation (9) to obtain the desired number of clusters according to the network parameters.

Step II: Randomly select one of the sensor nodes to be the first initial cluster center: $C_{1}$. Select another node located far away from $\;C_{1}$ to be the second cluster center: $C_{2}$ such that the distance between both the cluster centers is much greater than the transmission range $r_{t}$ of $C_{1}$. Similarly, select another node located far away from $C_{1}$ and $C_{2}$ to be the third cluster center: $C_{3}$ such that the distance between the cluster centers is the maximum of the min ($d\left(C_{3},C_{1}\right),d\left(C_{3},C_{2}\right))$ [32]. Select all the $C_{K}$ initial cluster centers in the mentioned sequence.

Step III: Calculate the distance between each sensor node and the initial cluster centers using the Euclidean distance given by Equation (10).
(10)$$d_{n-C}=\sqrt{\overset{n}{\underset{i=1}{\sum }}\left(S_{i}-S_{C}\right)}$$
where $d_{n-C}$ represents the distance of the sensor nodes to the cluster center, $S_{i}$ symbolizes the position coordinates of the *i*th node, and $S_{C}$ denotes the position coordinates of the cluster centers. Once the distances are calculated, each node is associated to the closest cluster center.

Step IV: Determine the new cluster centers by calculating the mean value of all the sensor nodes in respective clusters.

Step V: Repeat Step III with the new cluster centers. If the cluster centers assigned to the nodes change, then repeat Step IV, else stop the algorithm. These steps are summarized in Algorithm 1.

**Algorithm 1***K*-means clustering.1: **Begin**2: Calculate the value of $C_{K}$3: Set $C_{K}$ initial cluster centers according to step II.4: **For** (*i* = 1: number of cluster center) do5:  **For** (*j* = 1: number of nodes) do6:   $d_{n-C}=\sqrt{{\left(x_{S\_i}-x_{S\_C}\right)}^{2}+{\left(y_{S\_i}-y_{S\_C}\right)}^{2}}$
7:  **end**8: **end**
9: Assign each node to closest cluster center.10: **For** (*i* = 1:number of cluster center) do
11:  cluster centers (*i*) = mean(nodes assigned to cluster center (*i*))12: **end**13: **If** (previous cluster center != new cluster center) **then**14: repeat Algorithm 1 from line 4 to line 12.15: **else**16: stop17: **end**18: **End**

### 4.3. Cluster Head Selection Phase

After the network field is organized in the form of clusters, the nodes participate in the CH selection phase within each cluster. The objectives of the CH selection problem are as follows:

Organization of sensor nodes into several clusters, where each cluster has its own CH such that:
The node selected must have highest residual energy, i.e., $E_{i}\geq \;E_{std}$ as compared to other CM nodes of the particular cluster to enhance network lifetime.The CHs are well-distributed in the network to minimize energy consumption as it is directly proportional to the distance, i.e., $d_{i-C}\leq \;avg(d_{n-C})\;$(not concentrated in one area of the network) and serve their job for a long time period.

The transmission cost of some CMs may increase if the CH node is selected solely based on residual energy. It is also possible that the CH selected is located further away from some member nodes while being near to others. As a result, large amounts of communication energy were used by a member node to send its data to the selected CH, while the CH of another cluster was nearby. Our proposed scheme takes both the node’s residual energy and communication cost within the cluster into account for selecting a node to be a CH. During this phase, a standard weight function: $C_{std\_w}$ is calculated for each cluster using Equation (11) [17]:(11)$$C_{s\;td\_w}=\alpha \cdot E_{std}+\beta \cdot avg(d_{n-C})$$
where $E_{std}$ is the standard energy of a node that can participate in the CH selection process described in Section 3.2, $avg(d_{n-C})$ represents the average distance of all the CMs to the selected cluster center in the previous round, and α and β are algorithmic parameters ranging from (0,1), and these parameters should be balanced such that α+β=1.

Once the standard weight of each cluster is known, each node calculates its own weight according to Equation (12), and the node with weight closest to the standard weight of the corresponding cluster is selected as the CH for that round.
(12)$$C_{node\_w}=\alpha \cdot E_{i}+\beta \cdot d_{i-C}$$
where $E_{i}$ is the energy of *i*th node and $d_{i-C}$ represents the distance of the *i*th node to the selected cluster center. The CH selection process is presented in Algorithm 2. 

**Algorithm 2** CH selection.1: **Begin**2: **For** (*i* = 1: number of cluster centers) do3:   $C_{std\_w}=\alpha \cdot E_{std}+\beta \cdot avg(d_{n-C})$4:   **For** (*j* = 1: number of cluster members) do5:    $C_{node\_w}=\alpha \cdot E_{i}+\beta \cdot d_{i-C}$6:  **end**7: **end**8: **If** (weight of node ~= standard weight of the cluster) **then**
9:  select node as the CH for the corresponding cluster10: **else**11:  check other CM nodes with closest weight to $C_{std\_w}$12: **end**13: **End**

### 4.4. Data Transmission Phase

In the data transmission phase, the nodes stay idle until an event occurs. On detection of the event, the sensor node located in the close vicinity senses the information and transmits it to the CH using a multi-hop path. This multi-hop path selection is done on the basis of the fuzzy logic Dijkstra-based algorithm described as follows.

Suppose the nodes within a cluster along the CH are represented by a directed graph $D_{g}=\left(S,L\right),$ where *S* is the number of m sensor nodes in the cluster, i.e., $S_{i}\;\left(i=1,\;2,\;3,\ldots \ldots ,m\right)$ and *L* represents the set of links or edges between these nodes. Each edge $\left(S_{i},S_{j}\right)\in L$ has a weight such that $W\left(L_{ij}\right)$ is the weight of the link between the *i*th and the *j*th node. An edge $\left(S_{i},S_{j}\right)$ between the two nodes is defined to exist only if $S_{i}\;and\;S_{j}$ are within the radio transmission range of each other, i.e., if $d_{ij}\;\leq r_{t}$, where $d_{ij}$ is the distance between the *i*th and the *j*th node. The weight function for each link is influenced by two parameters: the residual energy of the neighbor node and distance of the neighbor node to the CH. This approach results in the increased energy efficiency of the network and eliminates the early death of the nodes due to node isolation. Once the weight for each neighboring node is computed, the node with the maximum energy and minimum intra-cluster communication cost, i.e., minimum weight, is selected as a data relay node through Dijkstra’s algorithm, as illustrated in Figure 4. 

Execution of the abovementioned process involved several steps described below:

Step I: Initialize set *SL* to store the neighbor nodes and corresponding links which will participate in the data transmission. Initially, the set *SL* consists of only the origin node $\left(S_{o}\right)$.
(13)$$SL=\left\{S_{o}\right\}$$

Step II: Identify the nodes located at the distance less than the transmission radius $r_{t}$ to the origin node $\left(S_{o}\right)$ as its neighbor nodes.

Step III: Determine the weight $W(L_{ij})$ of the link incident on each neighboring node by using Equation (14): (14)$$W(L_{ij})=\frac{1}{\gamma_{j}\cdot ICCC_{x}}$$
where $\gamma_{j}$ represents the fuzzy membership function [24] and $ICCC_{x}$ symbolizes the intra-cluster communication cost function [2] of cluster x. They are given by Equations (15) and (16), respectively, as: (15)$$\gamma_{j}=\{\begin{array}{cc} 1: & if\;E_{r}\left(S_{j}\right)=E_{init}\\ 1-\left(\frac{1-\mu }{1-\delta }\right).\left(1-\frac{E_{r}\left(S_{j}\right)}{E_{init}}\right): & \;if\;E_{th}\leq \;E_{r}\left(S_{j}\right)<E_{init}\\ \frac{\mu }{E_{th}-E_{TX}}.\left(E_{r}\left(S_{j}\right)-E_{TX}\right):\; & if\;E_{TX}\leq E_{r}\left(S_{j}\right)<E_{th}\\ 0:\; & if\;E_{r}\left(S_{j}\right)<E_{TX} \end{array}\}.$$
(16)$$ICCC_{x}=\frac{2.ASD_{CH}-d_{j}}{d_{j}}$$
where $E_{r}\left(S_{j}\right)$ represents the residual energy of the *j*th neighbor node, which is defined in Equation (17). Here, $E_{th}$ represents the threshold energy value computed as $\delta \cdot E_{init}$, where δ and μ are the algorithmic parameters, each ranging from (0, 1) [24]. Additionally, $ASD_{CH}$ is the average sensor node distance to the CH, and $d_{j}$ represents the distance of *j*th neighbor node to the CH.
(17)$$E_{r}\left(S_{j}\right)=E_{c}\left(S_{j}\right)-E_{TX}$$
where $E_{c}\left(S_{j}\right)$ represents the current energy of the *j*th neighbor node. The multiplicand part of Equation (14) ensures the next-hop node to be energy-rich, whereas the multiplier part assures the selection of a node that will reduce the communication cost within the cluster among other neighbor nodes.

Step IV: Compare the link weights of each neighbor node, select the link with the minimum weight value, and update set *SL* with the node and link information having the least link cost.

Step V: Repeat steps II to IV with the selected least link cost neighbor node until the final node is reached. The flowchart of the proposed algorithm is presented in Figure 5.

## 5. Performance Evaluation

In this section, the performance of the proposed algorithm was evaluated by MATLAB simulations and compared with FML, ESRAD, and SPFL algorithms which are defined in Section 2. In the simulations, the network domain was a square grid network layout of the size 500 × 500 m^2^ with 300 sensor nodes distributed randomly. The BS was located outside the network field at (400, 600). The BS was positioned outside the network to set up a more realistic environment. The values of µ and δ were selected to be 0.9 and 0.2 respectively because better results were obtained with these values [24]. The simulation parameters are listed in Table 2. Simulations were run for 500 iterations, and average values were considered to support the conclusion in this paper. 

### 5.1. Performance Metrics

To evaluate the performance of the proposed scheme, the following performance metrics were used:Lifetime of the networkNetwork energy conservation and consumptionDelayPercentage of packet lossNetwork throughput

To evaluate the performance of clustering routing protocols, the network lifetime was considered to be a significant index. There are several definitions of the network lifetime of a WSN. The time at which half of the network nodes are dead (HND) or when all the nodes in the network die (AND) can be referred to as the network lifetime index [33]. Additionally, the death of the first node (FND) in the network is an inflection point of the network, and it plays a vital role in the network lifetime analysis. Network energy conservation and consumption complement each other, and they are a major factor in reflecting the network performance. They can be defined as the proportion of the total amount of energy saved or used per node to transmit data to the BS. In this study, the packet delivery delay is defined as the physical distance between the nodes divided by the speed of an electromagnetic wave [34]. The percent packet loss metric represents that the relay nodes with energy less than the threshold level will not be able to participate in the data transmission process. It is defined in Equation (18). Throughput is the total number of data packets received at the BS per communication round [2].
(18)$$\mathrm{Packet}\;\mathrm{Loss}\;\left(\%\right)=\;\frac{\mathrm{Number}\;\mathrm{of}\;\mathrm{packets}\;\mathrm{transmitted}-\mathrm{number}\;\mathrm{of}\;\mathrm{packets}\;\mathrm{received}}{\mathrm{Number}\;\mathrm{of}\;\mathrm{packets}\;\mathrm{transmitted}}\times \;100$$

### 5.2. Results and Discussions

Figure 6a,b compares the network system lifetime of four routing schemes. When the number of rounds reaches 100, the gap among the algorithms for the number of dead nodes begins to be seen clearly. It can be noted that SPFL shows a drastic increase compared to other algorithms. It is owing to the fact that the PM located near the source node selects the next hop node to be the one with minimum distance from itself. This selection is done regardless of the energy level of the selected relay node, which results in the early death of nearby nodes.

Additionally, before the selection of the shortest path from the source to the destination node, the PM asks all the nodes in the network to send their information; this can cause unnecessary energy consumption of the nodes located far away from the PM. It can be seen that ESRAD and FML show a constantly increasing trend in terms of the number of dead nodes in the network. In the proposed scheme, the fuzzy membership function considers the residual energy of the neighbor nodes instead of the source node, as seen in FML and ESRAD. Moreover, the intra-cluster communication cost function selects a node located at the minimum distance to the CH among the neighbor nodes, which resulted in the even assignment of the data relaying tasks to each capable node. This behavior of the weight function encourages the energy conservation of the nodes and results in better performance compared to the other three algorithms. According to the results in Figure 6a, the data for certain specific number of rounds was extracted when some of the nodes died, and the percentage of the node death after each interval was plotted to visualize the performance comparison of the four algorithms more intuitively. It is evident from the results in Figure 6b that at round number 300, the percentage of dead nodes was almost 50 in SPFL, 30 in FML, 25 in ESRAD, and 20 in the proposed scheme. The comparison of performance of the existing algorithms with the proposed scheme reveals that the proposed scheme performs better owing to the consideration of the residual energies of neighbor nodes.

Figure 7a illustrates the comparison of the network energy conservation of the four routing schemes. The proposed scheme outperforms the other three routing algorithms owing to the consideration of the residual energy of the relay nodes. Additionally, selection of the node that offers minimum intra-cluster communication cost avoids the inclusion of the intermediate nodes that are located far from the CH, and therefore, saves the nodes from unnecessary depletion of energy owing to the longer distance as seen for FML and ESRAD. It is evident that SPFL has the lowest energy conservation and highest energy consumption because the PM proposes multi-hop path from the source to the destination node on the basis of shortest distance regardless of the energy of the intermediate nodes. Once the relay nodes located near the PM depleted their energies beyond a threshold level, the nodes positioned far away were being selected; this consumed extra power and resulted in minimum conservation of energy. Figure 7b presents the comparison of network energy consumption with respect to increasing number of nodes in the network. It can be observed from the graph that when network size is increased from 100 nodes to 500 nodes, the difference between the amounts of energy consumption for FML, ESRAD and proposed scheme is about 0.01 J whereas SPFL shows energy consumption of 0.1 J which is 1% of node’s total initial energy. This is because when a shortest path is required between a source and destination node, all the nodes send their information to the central PM node which results in unnecessary energy consumption within the network. Overall, the proposed routing scheme can provide better performance owing to the inclusion of residual energy and intra-cluster communication cost in the weight function while selecting the next hop for data transmission.

In Figure 8a, the system delay performance of the four routing schemes is compared with respect to the number of rounds. It can be inferred from the graph that until the 300th round, the SPFL shows better performance in terms of delay compared to proposed scheme; however, as the number of rounds increases, the delay of SPFL starts increasing even though it selects the shortest available path. This is owing to the reason that with the increase in the number of rounds, energy depletion of the nodes increases, and selecting the shortest path may not be the most efficient solution every time. Once the nodes located nearby deplete their energy, the faraway nodes are selected, which induces additional delay in the network. The system delay in case of ESRAD and FML is relatively high because both the schemes consider residual energies of the nodes, and they may select energy-rich nodes that are located far away, thereby increasing the delay of the whole network. Figure 8b illustrates the execution time of each algorithm with respect to the network size, i.e., increasing number of nodes. It is evident from the graph that FML and ESRAD have almost equal execution time as the algorithms focus on the residual energies of neighboring nodes to route data. In addition to this, SPFL shows relatively higher execution time when the network size is increased. This is because the PM asks all the nodes in the cluster to send their information for finding the shortest path towards the destination. When the number of nodes in the network increases, more information messages will be sent to the PM which will induce additional delays in the system, and hence increased execution time. The proposed scheme considers the residual energy of immediate nodes and its intra-cluster communication cost which leads to increased execution time compared to FML and ESRAD.

In Figure 9a, the percent packet loss comparison of the four routing protocols is performed to study the impact of packet loss on the system performance. In the 300 round, the packet loss for SPFL is 42%, which is double the loss for the proposed scheme, i.e., 21%. Moreover, ESRAD and FML show 5% and 12% more packet loss when compared with the proposed algorithm. This is owing to the fact that with an increase in the number of rounds, the nodes start depleting their energies faster. Furthermore, owing to the random distribution of the nodes, it is possible for a node to occasionally not find a neighbor node with sufficient energy or to find no intermediate node at all; this can result in packet loss. As illustrated, SPFL shows a greater percentage of packet loss because its only focus is to find the shortest path from the source to destination without considering the energy level of the data relaying nodes. However, ESRAD shows better performance compared to FML because it takes into account the energy consumption of the neighbor nodes in forwarding a packet, which makes it compulsory to choose a node with a sufficient amount of energy. The proposed scheme shows lower packet loss compared to the other three schemes because it balances the load distribution within the cluster by considering the node with reduced communication cost and high residual energy. Similarly, Figure 9b represents the number of packets received by the BS. In the proposed scheme, it is assumed that only one packet will be transmitted per round because it is an event-driven routing scheme that senses the event, and the nodes located near the event forward their data via multi-hop communication. It is evident that the proposed scheme has the highest number of packets transmitted to the BS when compared with the other three routing schemes. This is owing to the fact that the nodes die early in the other schemes and dissipate large amounts of energy which may cause the node isolation problem in some areas of the network, and the packets cannot reach the final node, which is CH in this study. Additionally, ESRAD shows better performance compared to FML and SPFL, where the latter shows the lowest packet reception at the BS. The SPFL scheme does not take into account the residual energies of the intermediate nodes causing packet loss owing to inefficient network energy consumption. Overall, the proposed scheme performs better owing to the consideration of neighbor node’s residual energy and distance to the CH, which improves the load balancing of the whole network and prolongs the average lifetime of each cluster node.

## 6. Conclusions

Wireless sensor networks require energy efficient schemes to enhance the network lifetime and system throughput, which provides minimum delay and improves the energy-efficiency of the overall system. In this study, an energy-efficient and delay tolerant routing scheme, which not only considers the residual energy of the neighbor node but also minimizes the intra-cluster communication cost of a cluster, was proposed. The proposed scheme divides the whole network into clusters and elects a CH for each cluster by using a weighted sum approach. Further, as soon as an event is triggered, the cluster nodes sense the event and starts the data transmission by using a route with minimum weight via Dijkstra’s routing algorithm. Simulation results indicate that compared with the FML, ESRAD, and SPFL algorithm, the proposed algorithm can prolong the network lifetime and improve the network energy conservation and consumption by reducing the energy consumption of individual nodes. Additionally, it shows delay tolerance owing to the consideration of the distance factor amidst data transmission. It also enhances the network throughput by reducing the packet loss and ensuring successful reception of the packets by the BS. As future work, the proposed scheme can be improved aiming at security routing problems to defend attacks from malicious adversary nodes in the network by establishing a trust evaluation model for each neighboring node to identify the most trust worthy next-hop node. In the data transmission, the links between the nodes and the CH can be made secure by taking into consideration the trust value of each node obtained from the evaluation model to increase the integrity of the received data, and the channel conditions can be considered to make it more suitable for practical applications. 

## Figures and Tables

**Figure 1 sensors-19-01040-f001:**
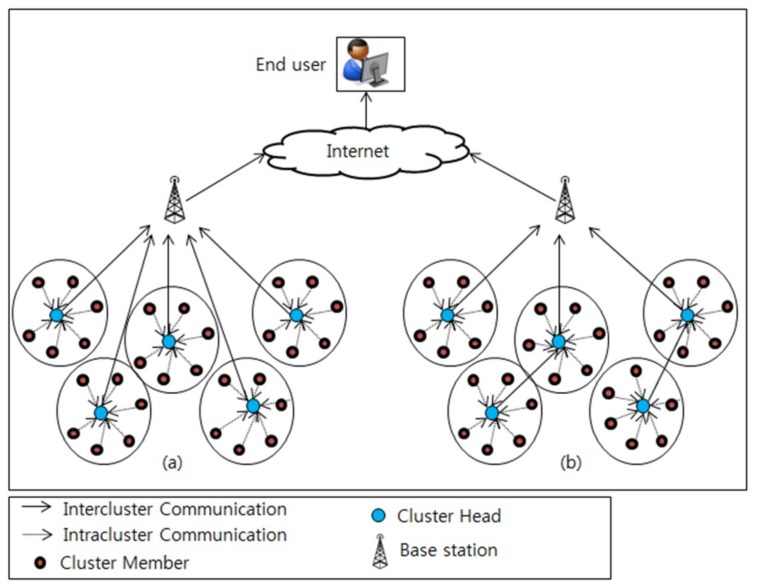
Cluster-based wireless sensor network (WSN) with different data communication scenarios: (**a**) single-hop communication and (**b**) multi-hop communication from sensor nodes to the end user through base station (BS).

**Figure 2 sensors-19-01040-f002:**
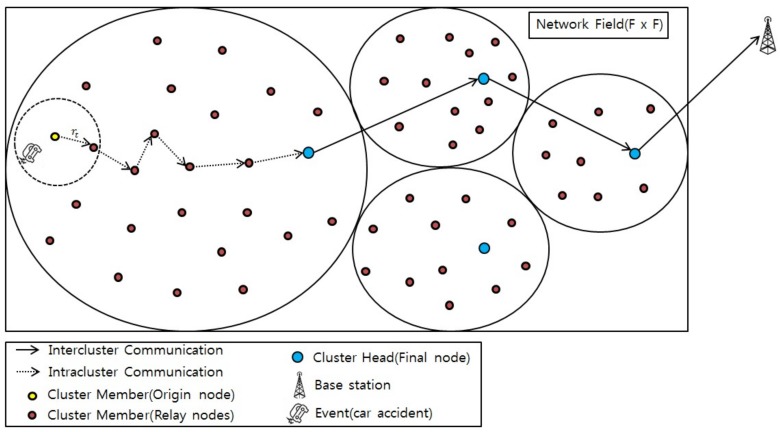
Network topology of proposed scheme.

**Figure 3 sensors-19-01040-f003:**
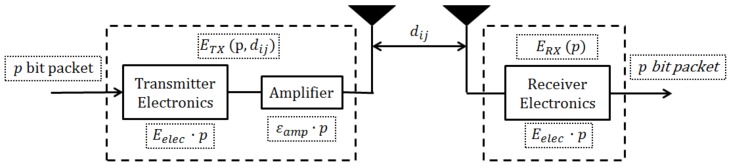
Radio energy dissipation model.

**Figure 4 sensors-19-01040-f004:**
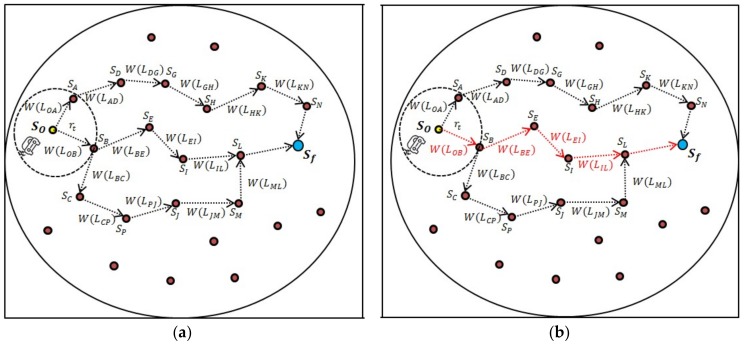
Multi-hop weighted path within a cluster: (**a**) directed graph with weights for each link between origin node ($S_{o})$ and destination node ($S_{f})$; (**b**) path with minimum weights selected from $S_{o}$ to $S_{f}$.

**Figure 5 sensors-19-01040-f005:**
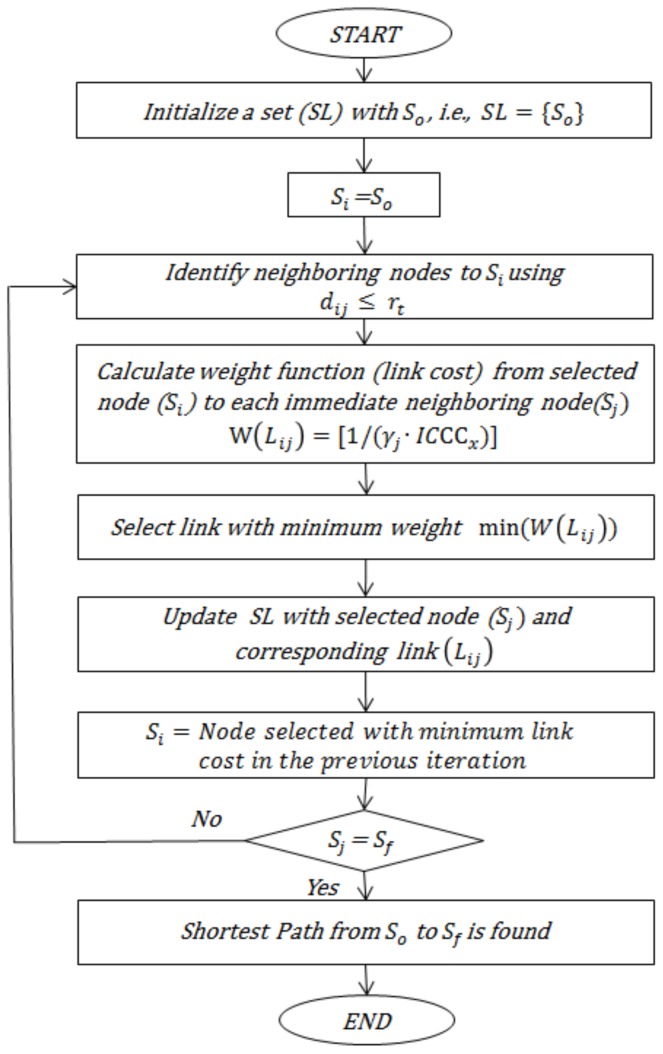
Flowchart of data transmission phase.

**Figure 6 sensors-19-01040-f006:**
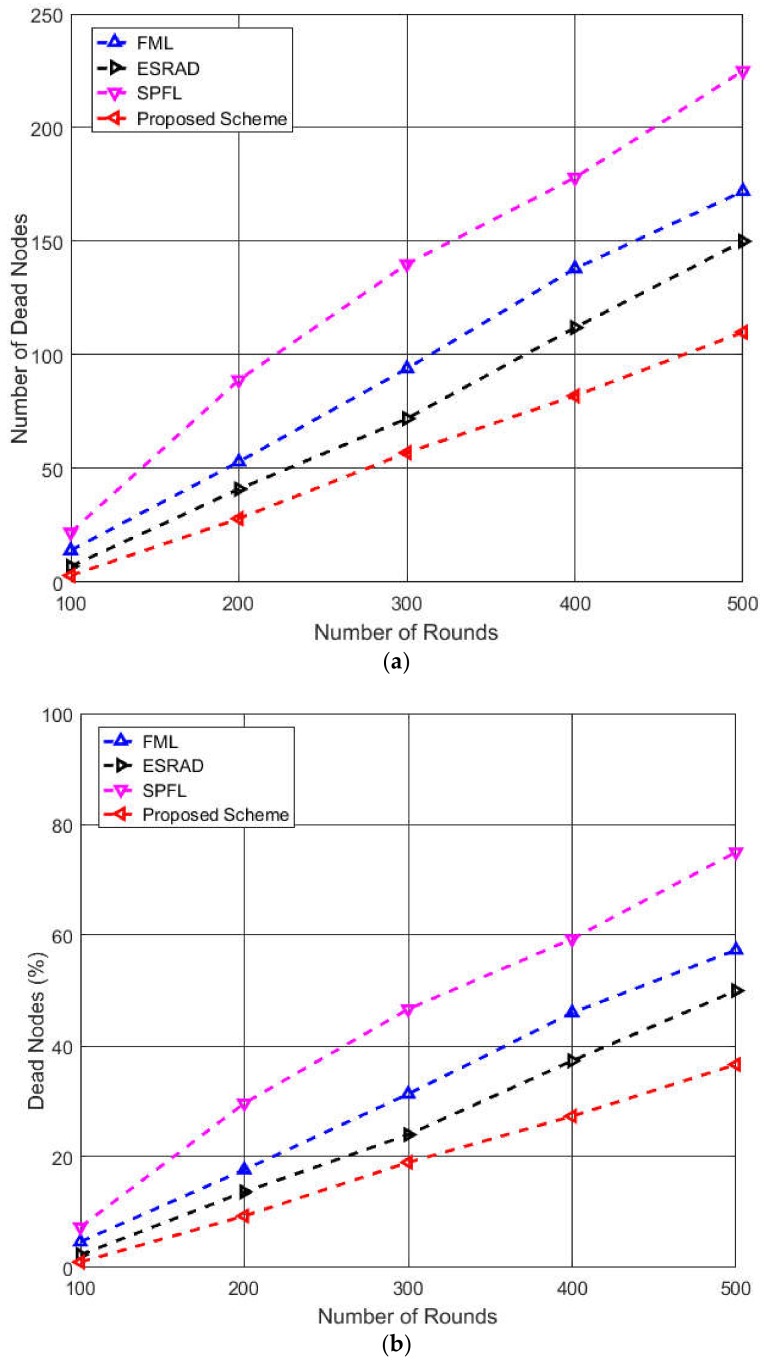
Performance comparison of network lifetime of the four algorithms: (**a**) number of dead nodes with respect to number of rounds; (**b**) percentage of dead nodes corresponding to specific number of rounds.

**Figure 7 sensors-19-01040-f007:**
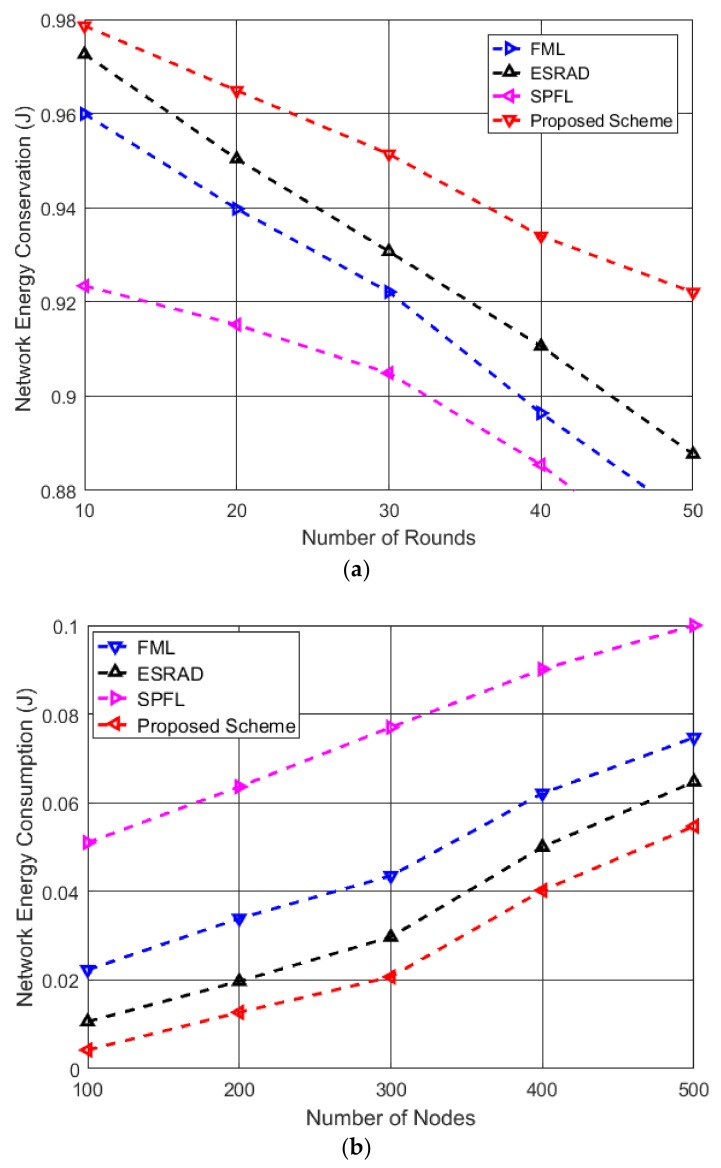
Performance comparison of network energy characteristics of the four algorithms: (**a**) network energy conservation with respect to number of rounds; (**b**) network energy consumption corresponding to number of nodes when the number of rounds is considered to be 100.

**Figure 8 sensors-19-01040-f008:**
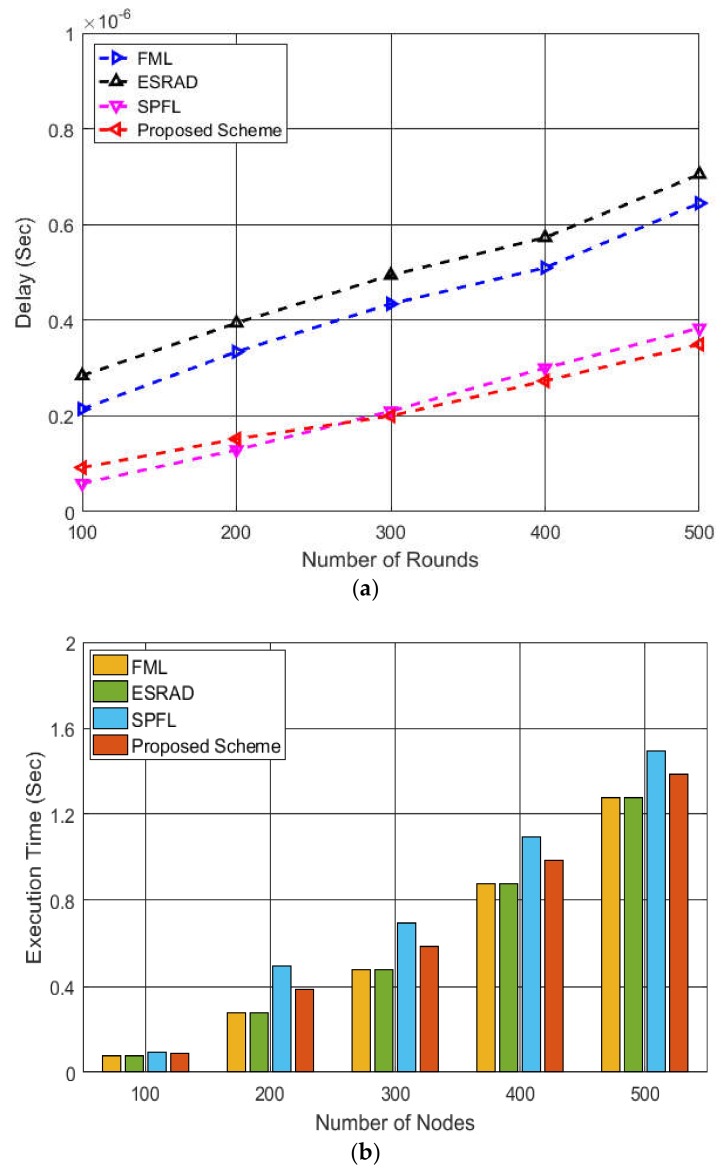
Performance comparison of the four routing algorithms: (**a**) system delay corresponding to specific number of rounds; (**b**) execution time of algorithms with respect to number of nodes.

**Figure 9 sensors-19-01040-f009:**
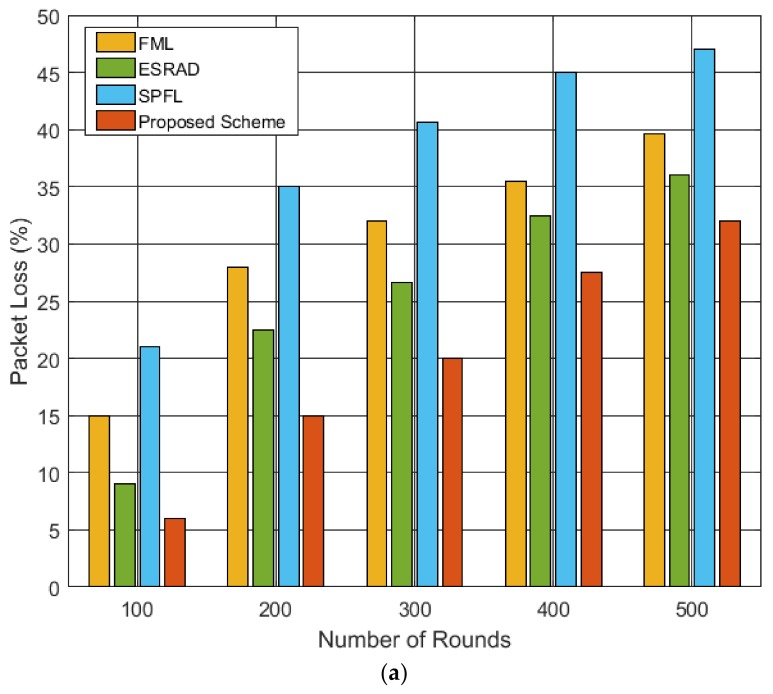
Performance comparison of routing schemes in terms of network throughput: (**a**) percent packet loss with respect to number of rounds; (**b**) number of packets sent to BS with respect to number of rounds.

**Table 1 sensors-19-01040-t001:** Key characteristics, advantages, and disadvantages of some cluster-based routing protocols.

Clustering Routing Protocol	Key Features	Advantages	Disadvantages
Low-energy adaptive clustering hierarchy (LEACH)	Introduced the concept of clustering and cluster head (CH) for wireless sensor network (WSN). Local compression to reduce global communication cost.	Localized coordination and control for cluster set-up and operation. Randomized rotation of cluster heads.	Inapplicable to time-constrained applications. Inapplicable to large scale due to single-hop communication.
Hybrid and energy-efficient distributed (HEED)	Equal-sized cluster formation. CH selection takes place on the basis of primary and secondary network parameters.	Minimizes intra-cluster communication energy consumption. Well-distributed CH nodes in the network.	Increased control messages overhead in cluster formation phase.
Cluster-based event-driven routing protocol (CERP)	Cluster formation takes place on the basis of various events. Distance-based link cost is calculated to compute shortest path.	Limits energy consumption by providing shortest distance between CH and base station.	Node isolation problem in the network. Unequal cluster formation due to event-occurrence.
Dijkstra-based weighted sum minimization (DWSM)	Multi-objective weighted function is calculated as link cost between nodes. Investigates the impact of varying weighting factor on wireless mesh network (WMN) performance.	Well-suited for time-constrained applications. Minimizes number of hops and provides good balance between path delay and capacity.	Inappropriate for energy-efficient scenarios.
Hierarchical unequal clustering fuzzy algorithm (HUCFA)	Network area is divided into three horizontal layers each split into grids. CH selection mechanism based on fuzzy logic.	Improves network lifetime as compared to LEACH.	Linguistic variables provide inefficient results when node mobility is increased.
Energy saving routing algorithm based on Dijkstra (ESRAD)	Calculates an evaluation index-based link cost to search path between two nodes. Dijkstra-algorithm is used to identify path with minimum energy consumption.	Minimizes energy consumption by accounting the energy dissipation in information transmission phase.	Residual energy of neighboring node is not considered while selecting next-hop. Inapplicable for real-time applications due to unpredictable delay.
Shortest path evaluation using fuzzy logic (SPFL)	Pool manager (PM) node is responsible for selecting shortest path. Fuzzy logic function is proposed for data routing in WSN.	Provides delay-efficient path from source to destination node.	Ignores the residual energies of the nodes in the network. Large overhead owing to control messages sent to PM node for finding least delay path.
Fuzzy maximum lifetime (FML)	Fuzzy membership function is used to calculate link weight. Minimum weighted path is selected via Dijkstra algorithm.	Maximizes network lifetime by taking the residual energy of source node into account.	Node mobility is not considered. Inapplicable to the time-constrained applications.
Distributed unequal clustering using fuzzy logic (DUCF)	Unequal clustering mechanism is proposed. Determines cluster-size and CH via fuzzy logic.	Balances energy consumption among clusters by forming appropriate sized clusters.	Neighbor node’s residual energy is not taken into account which may lead to load imbalance while data relaying.
Low-energy adaptive clustering hierarchy- dynamic threshold (LEACH-DT)	Dynamic energy threshold value is used to select CHs.	Resolves reallocation time slot problem among candidate and current CH nodes. Balances energy consumption of nodes in the network.	Uneven distribution of CHs may leads to hot-spot problem in the network.
Tree-cluster based shortest path (TCBSP)	Clustering approach of threshold sensitive energy efficient sensor network (TEEN) is used. Relay node optimization takes place by forming a tree-cluster structure.	Reduces transmission energy consumption in the network.	Large overhead due to multi-layer cluster formation.

**Table 2 sensors-19-01040-t002:** Simulation parameters.

Parameters	Values
Electronics Energy ($E_{elec}$)	50 nJ/bit
Data Aggregation Energy ($E_{DA}$)	5 nJ/bit/ signal
Initial Energy of Node ($E_{init}$)	1 J
Number of Nodes (*N*)	300
Position of BS (*X*,*Y*)	(400, 600)
Network Area (*F*)	500 × 500 m^2^
Packet Size (p)	1000 bits
CH to CH: Amplification Energy ($\;d\;<\;d_{th}$ )	$\varepsilon_{fs}=10\;\mathrm{pJ}/\mathrm{bit}/m^{2}$
CH to BS: Amplification Energy ($d\;\geq \;d_{th}$ )	$\varepsilon_{amp}$ = $0.0013\;\mathrm{pJ}/\mathrm{bit}/m^{4}$
μ, δ	0.9, 0.2
$T\mathrm{hreshold}\;\mathrm{Energy}\;(E_{th})$	0.2 J
Transmission Range of a Node ($r_{t})$	15 m

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
