# Peer review of "Fuzzy-Logic Dijkstra-Based Energy-Efficient Algorithm for Data Transmission in WSNs"

_sensors, 2019, doi:10.3390/s19051040_

Round 1
Reviewer 1 Report
The authors propose a new clustering mechanism for the wireless sensor networks towards improving the energy-efficient operation and the overall effectiveness of the considered wireless sensor network. The novelty of the proposed clustering mechanism lies on the selection of the cluster head that uses a weighted sum method to calculate the weight of each node in the cluster and compare it with the standard weight of that particular cluster. The topic of clustering the mobile nodes in the wireless sensor networks is not new in the literature, however, the proposed clustering method by the authors has the potential to improve the energy-efficiency in the operation of wireless sensor networks, thus, in the Internet of Things era.
The major comments that the authors should consider are the following.
1) Initially, the majority of the cited references are not recent, and the authors miss to refer to two main characteristics of the wireless sensor networks, i.e., mobility (e.g., "Routing in clustered multihop, mobile wireless networks with fading channel." In proceedings of IEEE SICON, vol. 97, no. 1997, pp. 197-211. 1997, "Adaptive clustering for mobile wireless networks." IEEE Journal on Selected areas in Communications 15, no. 7 (1997): 1265-1275) and social interest among the nodes towards exchanging information (e.g., "Interest, energy and physical-aware coalition formation and resource allocation in smart IoT applications." In Information Sciences and Systems (CISS), 2017 51st Annual Conference on, pp. 1-6. IEEE, 2017, "A socio-physical and mobility-aware coalition formation mechanism in public safety networks." EAI Endorsed Trans. Future Internet 4 (2018): 154176). Thus, Section 2 should be revised according to the previous comment.
2) Based on the reviewer’s understanding the proposed clustering mechanism is performed in a centralized manner and not distributed by a centralized entity. Within the considered network, which one is the aforementioned centralized entity that executes the clustering algorithm? Also, the authors should explain in the literature review the existing reinforcement learning techniques that have been proposed in order to realize clustering procedures (in the general area of networking environment) in a distributed and low-complexity manner (e.g., Reinforcement learning: An introduction. MIT press, 2018, "On the Problem of Optimal Cell Selection and Uplink Power Control in Open Access Multi-service Two-Tier Femtocell Networks." In International Conference on Ad-Hoc Networks and Wireless, pp. 114-127. Springer, Cham, 2014).
3) The authors compare their proposed clustering mechanism to other three clustering mechanisms from the recent literature. However, the authors have not provided the complexity analysis of their proposed clustering algorithm and also there are no supportive results to demonstrate if the proposed clustering algorithm operates in a real-time manner better compared the comparative scenarios. Towards this direction, some additional numerical results should be provided demonstrating: a) the execution time of the comparative algorithms and b) their scalability for increasing number of nodes in the wireless sensor network.
4) The authors provide a vague future work in Section 6, referring to security routing problems. However, the extension of the current research framework to secure routing is not obvious or even sketched by the authors. The authors should rewrite their provided future directions.
Author Response
We wish to express our appreciation to the Reviewers for their insightful comments, which
have helped us significantly to improve our manuscript. According to the suggestions, we
have thoroughly revised our manuscript and its revised version is enclosed. Point-by-point
responses to the comments is submitted.

Reviewer 2 Report
a new cluster head selection method that uses a weighted sum method to calculate the weight of each node in the cluster and compare it with the standard weight of that particular cluster is proposed in this paper. The node with weight closest to the standard cluster weight becomes the cluster head.
(1) Can you formulate the cluster head selection problem? What is the constraint?
(2) Section 2, a table summarizing the advantages and disadvantages of peer’s work is expected.
(3) Figure 2, what does “an event” mean?
(4) How fuzzy logic helps your method?
(5) Algorithm 1, why there are vertical bars on the left of the table?
(6) Equation 11, how do you balance the parameters alpha and beta?
(7) Some fuzzy papers could be discussed, see
a. Shuihua Wang, Detection of Dendritic Spines using Wavelet Packet Entropy and Fuzzy Support Vector Machine, CNS & Neurological Disorders - Drug Targets, 2017, 16(2): 116-121
b. Zhangjing Yang, Facial Emotion Recognition based on Biorthogonal Wavelet Entropy, Fuzzy Support Vector Machine, and Stratified Cross Validation, IEEE Access, 2016, 4: 8375-8385
Author Response
We wish to express our appreciation to the Reviewers for their insightful comments, which
have helped us significantly to improve our manuscript. According to the suggestions, we
have thoroughly revised our manuscript and its revised version is enclosed. Point-by-point
responses to the comments is provided.

Round 2
Reviewer 2 Report
Accept in current form.